# Social Hippocampus Memory Learning

Liping Yi [1]  Zhiming Zhao [1]  Kewen Zhu [1]  Xiang Li [1 2]  Zhiwei Shang [2]  Qinghua Hu [1]

## Abstract

Social learning highlights that learning agents improve not in isolation, but through interaction and structured knowledge exchange with others. When introduced into machine learning, this principle gives rise to *social machine learning* (SML), where multiple agents collaboratively learn by sharing abstracted knowledge. Federated learning (FL) provides a natural collaboration substrate for this paradigm, yet existing heterogeneous FL approaches often rely on sharing model parameters or intermediate representations, which may expose sensitive information and incur additional overhead. In this work, we propose **SoHip** (**So**cial **Hip**pocampus Memory Learning), a memory-centric social machine learning framework that enables collaboration among heterogeneous agents via memory sharing rather than model sharing. `SoHip` abstracts each agent's individual short-term memory from local representations, consolidates it into individual long-term memory through a hippocampus-inspired mechanism, and fuses it with collectively aggregated long-term memory to enhance local prediction. Throughout the process, raw data and local models remain on-device, while only lightweight memory are exchanged. We provide theoretical analysis on convergence and privacy preservation properties. Experiments on two benchmark datasets with seven baselines demonstrate that `SoHip` consistently outperforms existing methods, achieving up to 8.78% accuracy improvements. The code of `SoHip` is available at https://github.com/LipingYi/SoHip.

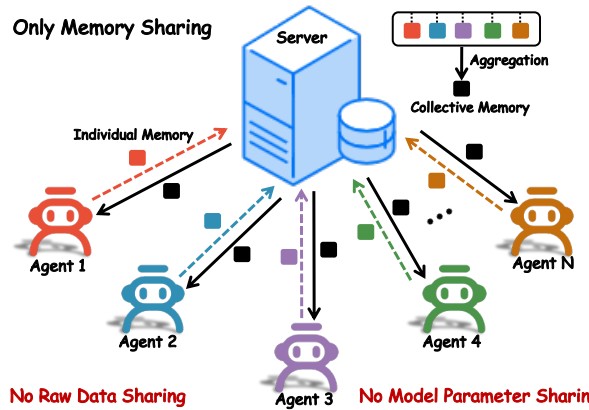

*Figure 1.* Memory-centric social machine learning framework.

## 1. Introduction

Social learning emphasizes that individuals do not learn in isolation by pure trial-and-error, but instead improve efficiently through interaction, observation, and information sharing with others. This idea can be traced back to Bandura's seminal work (Bandura, 1997), as stated below:

> *"Learning would be exceedingly laborious, not to mention hazardous, if people had to rely solely on the effects of their own actions to inform them what to do. Fortunately, most human behavior is learned observationally through modeling: from observing others one forms an idea of how new behaviors are performed, and on later occasions this coded information serves as a guide for action."* — *Bandura (1977)*

These observations suggest that abstracting experience, storing knowledge, and sharing memory are central mechanisms underlying effective social learning.

When introduced into machine learning, this principle naturally leads to the paradigm of *social machine learning* (SML) (Yao et al., 2024), where multiple learning entities (commonly modeled as autonomous agents) collaborate and improve collectively through structured knowledge exchange. A variety of classical collective learning approaches, including ant colony optimization (Dorigo et al., 2007), bee colony algorithms (Karaboga & Akay, 2009), ensemble learning (Sagi & Rokach, 2018), and federated learning (McMahan et al., 2017), can be viewed as concrete instantiations of social machine learning.

[1] School of Artificial Intelligence, Tianjin University, Tianjin, China [2] School of Data Science, The Chinese University of Hong Kong, Shenzhen, China. Correspondence to: Qinghua Hu <huqinghua@tju.edu.cn>.

Among them, federated learning (FL) (Kairouz et al., 2021; Yang et al., 2019) has received particular attention due to its ability to enable collaborative training without sharing raw data, making it especially suitable for privacy-sensitive applications such as financial risk control (Cheng et al., 2020) and medical diagnosis (Rauniyar et al., 2023; Huang et al., 2026). In this work, we adopt FL as the underlying collaboration substrate and further explore how social learning properties can be systematically incorporated into privacy-constrained collaborative learning.

FL coordinates multiple distributed agents through a central server to achieve collective optimization without exposing local data. Despite its success, FL faces several fundamental challenges in real-world deployments. Local data across agents are often highly non-independent and non-identically distributed (non-IID) (Zhu et al., 2021; Tan et al., 2022a; Meng et al., 2024; Qi et al., 2025; Kou et al., 2025; 2026); system capabilities such as communication, computation, and storage are heterogeneous (Horváth, 2021; Diao, 2021); and in many scenarios, agents maintain inherently heterogeneous model architectures (Ye et al., 2024; Yi et al., 2023; Li et al., 2026; Han et al., 2025). To address these issues, existing heterogeneous FL approaches typically rely on sharing aligned subsets of model parameters (Collins et al., 2021; Liang et al., 2020), exchanging intermediate representations (Tan et al., 2022b) or model outputs (Jeong et al., 2018), or introducing additional homogeneous auxiliary models as knowledge carriers (Shen et al., 2020; Wu et al., 2022). Although effective to some extent, these strategies may still expose sensitive model- or data-related information and often incur non-negligible computational and communication overhead.

To enable efficient and privacy-friendly collaboration among agents with heterogeneous models, we revisit knowledge sharing in FL from a social learning perspective. In social learning, individuals do not directly replicate others' behaviors or decisions; instead, they abstract, store, and integrate others' experiences into reusable internal knowledge representations. Motivated by this observation, we propose **SoHip** (**So**cial **Hip**pocampus Memory Learning), a memory-centric social machine learning framework built upon federated collaboration.

SoHip introduces memory as the primary carrier of social knowledge exchange, shown as Fig. 1. (1) Specifically, each agent first extracts representations using its local heterogeneous model and forms *individual short-term memory* through a short-term memory abstraction module. (2) Inspired by the role of the hippocampus 🦛 in consolidating short-term experiences into long-term memory, SoHip integrates individual short-term memory with historical individual long-term memory via a hippocampus-inspired short-to-long memory conversion module, thereby updating

individual long-term memory. (3) The updated individual long-term memory is then fused with the *collective long-term memory* received from the server through an individual–collective memory fusion module, yielding a complete memory representation that enhances local prediction. (4) After local training, each agent uploads its updated individual long-term memory to the server, where collective aggregation produces a new collective long-term memory that is broadcast in the next communication round. Throughout the entire SoHip workflow, raw data and local model parameters remain strictly on-device; only highly abstracted memory are exchanged, enabling effective collaboration while preserving both data and model privacy.

The main contributions are summarized as follows:

- We propose SoHip, a novel social machine learning framework that introduces memory as a social knowledge-sharing carrier, enabling collaborative learning across data, system, and model heterogeneity without sharing raw data or model parameters.

- We provide theoretical analysis on the convergence behavior and privacy preservation properties of the proposed framework, offering principled guarantees for its effectiveness.

- Extensive experiments on two benchmark datasets against seven representative baselines demonstrate that SoHip consistently achieves superior performance, yielding up to 8.78% accuracy improvements.

## 2. Related Work

### 2.1. Social Learning and Social Machine Learning

Social learning (Bandura, 1997) originates from behavioral and cognitive science and emphasizes that individuals improve their behavior not only through isolated trial-and-error, but also by observing others, interacting with peers, and sharing accumulated experience. Rooted in social learning theory, this perspective highlights the central roles of experience abstraction, memory formation, and knowledge reuse in efficient learning processes.

When these principles are introduced into machine learning systems, they give rise to *social machine learning* (Yao et al., 2024), where multiple learning agents collaboratively improve through structured information exchange. Representative paradigms under this umbrella include swarm intelligence methods such as ant colony (Dorigo et al., 2007) and bee colony optimization (Karaboga & Akay, 2009), ensemble learning (Sagi & Rokach, 2018), and federated learning (McMahan et al., 2017). In these approaches, learning agents benefit from shared or aggregated knowledge to achieve improved group-level performance.

Despite their success, most existing social machine learning methods either assume homogeneous models or rely on tightly coupled interaction mechanisms, which limits their applicability in heterogeneous and privacy-constrained environments. In contrast, `SoHip` instantiates social machine learning from a memory-centric perspective, enabling heterogeneous agents to interact via abstracted memory without exposing raw data or local model parameters.

## 2.2. Heterogeneous Federated Learning

Federated learning enables collaborative model training across distributed clients without sharing raw data, and has become a prominent paradigm for privacy-preserving collaborative learning (Yang et al., 2019; Kairouz et al., 2021). In practical deployments, however, clients often exhibit significant heterogeneity in data distributions (Zhu et al., 2021; Tan et al., 2022a), system resources (Horváth, 2021; Diao, 2021; Yi et al., 2022; 2024b;a), and model architectures (Ye et al., 2024).

To address these challenges, existing heterogeneous federated learning approaches typically rely on three representative strategies: (i) sharing aligned homogeneous subsets of model parameters by decoupling local models into heterogeneous and homogeneous components, so that only homogeneous parameters are aggregated across clients (Liang et al., 2020; Chen et al., 2021; Collins et al., 2021; Oh et al., 2022; Pillutla et al., 2022; Jang et al., 2022; Liu et al., 2022; Yi et al., 2023); (ii) exchanging intermediate representations or prediction outputs to transfer task-relevant information while avoiding direct parameter sharing (*i.e.*, `FedProto` (Tan et al., 2022b), `FedSSA` (Yi et al., 2024c), `FedRAL` (Yi et al., 2025a) and others (Jeong et al., 2018; Ahn et al., 2019; 2020; He et al., 2020)); and (iii) introducing auxiliary homogeneous models shared across clients to serve as intermediaries for knowledge transfer between heterogeneous local models, suhc as `FedKD` (Wu et al., 2022), `FedMRL` (Yi et al., 2024d), `pFedES` (Yi et al., 2025b) and others (Shen et al., 2020; Kalra et al., 2023; Qin et al., 2023; Yi et al., 2026a;b). While effective in certain scenarios, these strategies may still expose partial model behavior or incur additional computation and communication overhead, which can limit scalability and privacy guarantees.

In contrast to prior work, `SoHip` revisits collaborative learning from a social learning perspective and introduces *memory* as the primary carrier of social knowledge exchange. Rather than sharing model parameters, intermediate features, or predictions, each agent abstracts local experience into short-term memory, consolidates it into long-term memory via a hippocampus-inspired mechanism, and exchanges only compact long-term memory with the server for collective aggregation. By decoupling knowledge sharing from model structure and data semantics, `SoHip` enables effective collaboration across heterogeneous agents while preserving both data privacy and model autonomy, providing a memory-centric view of social machine learning under heterogeneity and privacy constraints.

## 3. Problem Definition

We consider a *social machine learning* problem involving $N$ distributed learning agents, each associated with a private local dataset and a potentially heterogeneous model. Agent $i$ holds a local dataset $\mathcal{D}_i$ and maintains a feature extractor $\mathcal{F}_i$ together with a local classifier $\mathcal{H}_i$. The local data distributions $\{\mathcal{D}_i\}_{i=1}^N$ are generally non-IID, and the local model architectures $\{\mathcal{F}_i, \mathcal{H}_i\}$ may differ across agents.

The agents aim to collaboratively improve their predictive performance by leveraging experience from others, while satisfying the following fundamental constraints: (1) raw local data must remain strictly on-device; (2) local model parameters are not directly shared across agents; and (3) collaboration must be robust to data, system, and model heterogeneity. Such constraints naturally arise in privacy-sensitive and resource-heterogeneous environments, and preclude direct parameter or representation sharing.

From a social learning perspective, we view collaboration as a process of *memory-based knowledge exchange*. Rather than sharing model parameters, intermediate features, or prediction outputs, we assume that each agent maintains an internal *memory state* that abstracts and stores its accumulated experience. Specifically, at communication round $t$, agent $i$ maintains an *individual memory* $\mathbf{M}_i^t \in \mathbb{R}^m$, where $m$ denotes a shared memory dimension. A central server maintains a *collective memory* $\mathbf{M}^t \in \mathbb{R}^m$, which aggregates individual memories and serves as a shared repository of group-level knowledge.

The objective of social machine learning in this setting is to improve the local prediction performance of each heterogeneous agent $i$ through memory exchange:

$$\min_{\{\mathcal{F}_i, \mathcal{H}_i\}} \sum_{i=1}^N \mathbb{E}_{(\mathbf{x},y)\sim\mathcal{D}_i}\big[\ell\big(\mathcal{H}_i(\mathcal{F}_i(\mathbf{x})),\, y\big)\big], \qquad (1)$$

subject to the constraint that cross-agent interaction is conducted *exclusively* through memory $\{\mathbf{M}_i^t\}_{i=1}^N$ and $\mathbf{M}^t$.

The central challenge addressed in this work is therefore: *How can agents with heterogeneous models effectively abstract, consolidate, and exchange memory to enable social machine learning, while preserving data and model privacy and remaining robust to heterogeneity?* In the following section, we introduce `SoHip`, a memory-centric framework that provides a principled solution to this challenge. [1]

---

[1]The key notations are summarized in Appendix A.

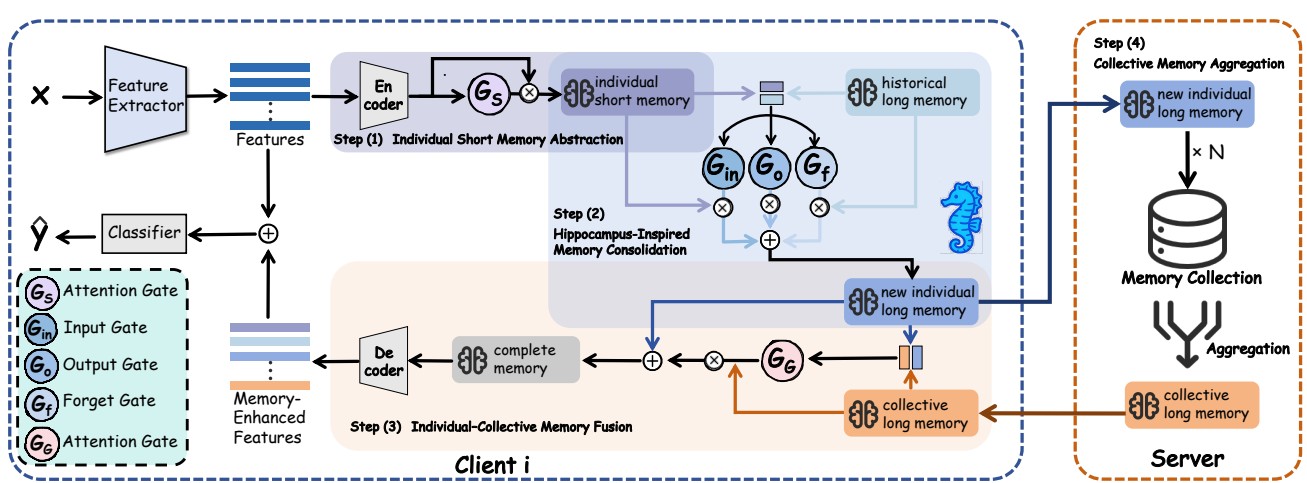

*Figure 2.* Overview of SoHip. SoHip operates sequentially by (1) abstracting *individual short-term memory* from local representations, (2) consolidating it into *individual long-term memory* via a hippocampus-inspired mechanism, (3) fusing it with *collective long-term memory* for enhanced prediction, and (4) aggregating updated individual long-term memories to form an updated collective memory.

## 4. The Proposed SoHip Algorithm

We present SoHip, a memory-centric social machine learning framework designed to enable collaboration among heterogeneous agents without sharing raw data or local model parameters. Instead of exchanging parameters or intermediate representations, SoHip introduces *memory* as the primary carrier of social knowledge. As illustrated in Figure 2, the framework consists of four functional modules that progressively abstract, consolidate, and exchange memory across agents. The complete SoHip algorithm is described in Alg. 1.

### 4.1. Individual Short-Term Memory Abstraction

In social learning, individuals do not retain all raw experiences; instead, recent observations are selectively abstracted into compact short-term memory, where salient information is emphasized and redundant or noisy signals are suppressed. Following this principle, each agent first extracts latent representations from its private data using a heterogeneous feature extractor.

Given a mini-batch $\mathcal{B}_i^t$, agent $i$ computes

$$\mathbf{Z}_i^t = \mathcal{F}_i(\mathcal{B}_i^t), \quad \mathbf{Z}_i^t \in \mathbb{R}^{B_i^t \times d_i}, \tag{2}$$

where $B_i^t$ denotes the batch size and $d_i$ is the feature dimension of agent $i$. To abstract recent experience, the representations are projected into a shared memory space via a lightweight encoder (one linear layer):

$$\mathbf{Z}_{i,\text{enc}}^t = \mathcal{E}_i(\mathbf{Z}_i^t), \quad \mathbf{Z}_{i,\text{enc}}^t \in \mathbb{R}^{B_i^t \times m}, \tag{3}$$

where $m \leq d_i$ is the shared memory dimension. Batch-level information is summarized by averaging along the batch dimension:

$$\bar{\mathbf{z}}_i^t = \frac{1}{B_i^t} \sum_{b=1}^{B_i^t} \mathbf{Z}_{i,\text{enc},b}^t \in \mathbb{R}^m. \tag{4}$$

Rather than directly storing this summary, SoHip introduces a lightweight gating unit to assess the *importance* of the current observations. The gate acts as an adaptive filter, highlighting informative dimensions while attenuating less relevant or noisy signals:

$$\boldsymbol{\alpha}_i^{\text{S},t} = \sigma\left(\mathcal{G}_\text{S}(\bar{\mathbf{z}}_i^t)\right), \tag{5}$$

where $\mathcal{G}_\text{S}(\cdot)$ is a lightweight gating unit implemented as a single linear layer, and $\sigma(\cdot)$ denotes the sigmoid activation that produces dimension-wise importance scores in $(0, 1)$.

The resulting individual short-term memory is defined as

$$\mathbf{M}_i^{\text{S},t} = \boldsymbol{\alpha}_i^{\text{S},t} \cdot \bar{\mathbf{z}}_i^t, \tag{6}$$

which encodes a compact and selectively weighted representation of the agent's recent experience.

### 4.2. Hippocampus-Inspired Memory Consolidation

In human cognition, the hippocampus plays a critical role in consolidating short-term experiences into long-term memory by selectively integrating new information while preserving previously acquired knowledge. Inspired by this biological mechanism, SoHip updates individual long-term memory through a gated short-to-long memory consolidation process.

Specifically, the newly formed short-term memory $\mathbf{M}_i^{\text{S},t}$ and the historical long-term memory $\mathbf{M}_i^{\text{L},t-1}$ are first concatenated as

$$\mathbf{u}_i^t = \left[\mathbf{M}_i^{\text{S},t}; \mathbf{M}_i^{\text{L},t-1}\right]. \tag{7}$$

Based on this combined representation, three gating units are employed to regulate memory consolidation:

$$\boldsymbol{\alpha}_i^{\text{in},t} = \sigma\big(\mathcal{G}_{\text{in}}(\mathbf{u}_i^t)\big), \boldsymbol{\alpha}_i^{\text{f},t} = \sigma\big(\mathcal{G}_{\text{f}}(\mathbf{u}_i^t)\big), \boldsymbol{\alpha}_i^{\text{o},t} = \sigma\big(\mathcal{G}_{\text{o}}(\mathbf{u}_i^t)\big),$$
$$\tag{8}$$

where $\mathcal{G}_{\text{in}}(\cdot)$, $\mathcal{G}_{\text{f}}(\cdot)$, and $\mathcal{G}_{\text{o}}(\cdot)$ are lightweight gating units implemented as single linear layers, and $\sigma(\cdot)$ denotes the sigmoid activation. The input gate $\boldsymbol{\alpha}_i^{\text{in},t}$ controls how much newly abstracted short-term memory should be incorporated, the forget gate $\boldsymbol{\alpha}_i^{\text{f},t}$ regulates the retention of historical long-term memory, and the output gate $\boldsymbol{\alpha}_i^{\text{o},t}$ modulates the overall strength of the consolidated memory.

The updated individual long-term memory is then computed as

$$\mathbf{M}_i^{\text{L},t} = \boldsymbol{\alpha}_i^{\text{o},t}\Big(\boldsymbol{\alpha}_i^{\text{in},t} \cdot \mathbf{M}_i^{\text{S},t} + \boldsymbol{\alpha}_i^{\text{f},t} \cdot \mathbf{M}_i^{\text{L},t-1}\Big). \tag{9}$$

Through this gated consolidation mechanism, each agent selectively integrates informative new experience while preserving stable historical knowledge, thereby enabling robust and continual memory accumulation under non-IID data and model heterogeneity.

### 4.3. Individual–Collective Memory Fusion

Beyond consolidating individual experience, effective social learning further requires each agent to selectively absorb useful collective knowledge that complements its local understanding. After updating individual long-term memory, SoHip integrates it with the collective long-term memory aggregated in the previous round and broadcast to agents.

Specifically, the updated individual long-term memory $\mathbf{M}_i^{\text{L},t}$ and the received collective long-term memory $\mathbf{M}^{\text{L},t-1}$ are concatenated to form the fusion input:

$$\mathbf{v}_i^t = \big[\mathbf{M}_i^{\text{L},t}; \mathbf{M}^{\text{L},t-1}\big]. \tag{10}$$

A fusion gating unit is then applied to determine which components of the collective memory are beneficial to the local agent:

$$\boldsymbol{\alpha}_i^{\text{G},t} = \sigma\big(\mathcal{G}_{\text{G}}(\mathbf{v}_i^t)\big), \tag{11}$$

where $\mathcal{G}_{\text{G}}(\cdot)$ is implemented as a single linear layer, and $\sigma(\cdot)$ denotes the sigmoid activation. The resulting gate $\boldsymbol{\alpha}_i^{\text{G},t}$ assigns dimension-wise importance scores, enabling each agent to selectively absorb the shared collective knowledge relevant to its local context.

The complete memory is then constructed as

$$\mathbf{M}_i^t = \boldsymbol{\alpha}_i^{\text{G},t} \cdot \mathbf{M}^{\text{L},t-1} + \mathbf{M}_i^{\text{L},t}. \tag{12}$$

To enhance local prediction, the complete memory is projected back to the original feature space via a lightweight decoder (one linear layer):

$$\tilde{\mathbf{m}}_i^t = \mathcal{R}_i(\mathbf{M}_i^t), \quad \tilde{\mathbf{m}}_i^t \in \mathbb{R}^{d_i}. \tag{13}$$

The decoded memory is expanded along the batch dimension and combined with the original representations through a residual connection:

$$\hat{\mathbf{Z}}_i^t = \mathbf{Z}_i^t + \text{Expand}(\tilde{\mathbf{m}}_i^t). \tag{14}$$

Finally, predictions are obtained as

$$\hat{\mathbf{Y}}_i^t = \mathcal{H}_i(\hat{\mathbf{Z}}_i^t). \tag{15}$$

### 4.4. Collective Memory Aggregation

At the group level, SoHip accumulates social knowledge through collective memory aggregation. After local consolidation, each participating agent uploads its updated individual long-term memory $\mathbf{M}_i^{\text{L},t}$ to the server.

The server aggregates the received memories to form the collective long-term memory for the next round:

$$\mathbf{M}^{\text{L},t+1} = \sum_{i \in \mathcal{S}_t} p_i \, \mathbf{M}_i^{\text{L},t}, \tag{16}$$

where $\mathcal{S}_t$ denotes the set of participating agents and $p_i$ is the aggregation weight (*e.g.*, proportional to local data size).

Unlike conventional parameter aggregation, this operation aggregates highly abstracted long-term memory, which encapsulates distilled experience from heterogeneous agents. The resulting collective memory serves as a shared repository of social knowledge and is broadcast to agents in the next round, where it is selectively absorbed via the individual–collective memory fusion module. Through iterative aggregation and selective absorption, SoHip enables continual refinement of collective experience across heterogeneous agents while preserving data and model privacy.

## 5. Theoretical Analysis

We analyze the convergence behavior and privacy properties of SoHip. Our analysis shows that memory-based social machine learning preserves the convergence guarantees of federated optimization, while preventing direct leakage of local data or model parameters.

**Theorem 5.1** (Convergence of SoHip). *Assume that each local objective $f_i$ is L-smooth and stochastic gradients have bounded variance. Under a suitable stepsize, the sequence generated by* SoHip *satisfies*

$$\frac{1}{T}\sum_{t=0}^{T-1} \mathbb{E}\big[\|\nabla f(\theta^t)\|^2\big] = \mathcal{O}\left(\frac{1}{\sqrt{T}}\right) + \mathcal{O}(\Delta_{\text{het}}),$$

*where $f(\theta) = \sum_{i=1}^{N} p_i f_i(\theta)$ and $\Delta_{\text{het}}$ characterizes the effect of data and model heterogeneity across agents.*

*Discussion.* The above result indicates that introducing gated memory abstraction, hippocampus-inspired consolidation, and individual–collective memory fusion does not

**Algorithm 1** `SoHip`

---

1: **Input:** Agents $\{(\mathcal{D}_i, \mathcal{F}_i, \mathcal{H}_i, \mathcal{E}_i, \mathcal{R}_i)\}_{i=1}^N$; memory dimension $m$; participation rate $C$; aggregation weights $\{p_i\}$; initial collective memory $\mathbf{M}^{L,0} \in \mathbb{R}^m$; initial individual long-term memories $\{\mathbf{M}_i^{L,0} \in \mathbb{R}^m\}$.
2: **for** round $t = 1, 2, \ldots, T$ **do**
3:     Server samples participating set $\mathcal{S}_t$ with $|\mathcal{S}_t| = \lfloor CN \rfloor$ and broadcasts $\mathbf{M}^{L,t-1}$.
4:     **for all** agent $i \in \mathcal{S}_t$ **in parallel do**
5:         Sample mini-batch $\mathcal{B}_i^t$ from $\mathcal{D}_i$.
6:         **(I) Individual short-term memory abstraction.**
7:         $\mathbf{Z}_i^t \leftarrow \mathcal{F}_i(\mathcal{B}_i^t)$                 // $\mathbf{Z}_i^t \in \mathbb{R}^{B_i^t \times d_i}$
8:         $\mathbf{Z}_{i,\text{enc}}^t \leftarrow \mathcal{E}_i(\mathbf{Z}_i^t)$         // $\mathbf{Z}_{i,\text{enc}}^t \in \mathbb{R}^{B_i^t \times m}$
9:         $\bar{\mathbf{z}}_i^t \leftarrow \frac{1}{B_i^t} \sum_{b=1}^{B_i^t} \mathbf{Z}_{i,\text{enc},b}^t$     // $\bar{\mathbf{z}}_i^t \in \mathbb{R}^m$
10:        $\boldsymbol{\alpha}_i^{S,t} \leftarrow \sigma(\mathcal{G}_S(\bar{\mathbf{z}}_i^t))$          // $\boldsymbol{\alpha}_i^{S,t} \in (0,1)^m$
11:        $\mathbf{M}_i^{S,t} \leftarrow \boldsymbol{\alpha}_i^{S,t} \odot \bar{\mathbf{z}}_i^t$         // $\mathbf{M}_i^{S,t} \in \mathbb{R}^m$
12:        **(II) Hippocampus-inspired memory consolidation.**
13:        $\mathbf{u}_i^t \leftarrow [\mathbf{M}_i^{S,t}; \mathbf{M}_i^{L,t-1}]$
14:        $\boldsymbol{\alpha}_i^{\text{in},t} \leftarrow \sigma(\mathcal{G}_{\text{in}}(\mathbf{u}_i^t)), \quad \boldsymbol{\alpha}_i^{\text{f},t} \leftarrow \sigma(\mathcal{G}_{\text{f}}(\mathbf{u}_i^t)),$
15:        $\boldsymbol{\alpha}_i^{\text{o},t} \leftarrow \sigma(\mathcal{G}_{\text{o}}(\mathbf{u}_i^t))$
16:        $\mathbf{M}_i^{L,t} \leftarrow \boldsymbol{\alpha}_i^{\text{o},t} \odot (\boldsymbol{\alpha}_i^{\text{in},t} \odot \mathbf{M}_i^{S,t} + \boldsymbol{\alpha}_i^{\text{f},t} \odot \mathbf{M}_i^{L,t-1})$
17:        **(III) Individual–collective memory fusion.**
18:        $\mathbf{v}_i^t \leftarrow [\mathbf{M}_i^{L,t}; \mathbf{M}^{L,t-1}]$
19:        $\boldsymbol{\alpha}_i^{G,t} \leftarrow \sigma(\mathcal{G}_G(\mathbf{v}_i^t))$
20:        $\mathbf{M}_i^t \leftarrow \boldsymbol{\alpha}_i^{G,t} \odot \mathbf{M}^{L,t-1} + \mathbf{M}_i^{L,t}$
21:        $\tilde{\mathbf{m}}_i^t \leftarrow \mathcal{R}_i(\mathbf{M}_i^t)$          // $\tilde{\mathbf{m}}_i^t \in \mathbb{R}^{d_i}$
22:        $\hat{\mathbf{Z}}_i^t \leftarrow \mathbf{Z}_i^t + \text{Expand}(\tilde{\mathbf{m}}_i^t)$    // $\hat{\mathbf{Z}}_i^t \in \mathbb{R}^{B_i^t \times d_i}$
23:        $\hat{\mathbf{Y}}_i^t \leftarrow \mathcal{H}_i(\hat{\mathbf{Z}}_i^t)$
24:        Update local parameters of $\mathcal{F}_i, \mathcal{H}_i, \mathcal{E}_i, \mathcal{R}_i$ and gating units by minimizing the local loss on $\mathcal{B}_i^t$.
25:        Upload $\mathbf{M}_i^{L,t}$ to the server.
26:     **end for**
27:     **(IV) Collective memory aggregation.**
28:     $\mathbf{M}^{L,t} \leftarrow \sum_{i \in \mathcal{S}_t} p_i \mathbf{M}_i^{L,t}$.
29: **end for**
30: **Output:** Final local heterogeneous models $\{(\mathcal{F}_i, \mathcal{H}_i)\}_{i=1}^N$.

---

hinder convergence. The additional heterogeneity term $\Delta_{\text{het}}$ is unavoidable in heterogeneous settings and is empirically mitigated by memory-based knowledge sharing.

**Theorem 5.2** (Privacy Preservation). *During training,* `SoHip` *never transmits raw data, local model parameters, intermediate features, or prediction outputs. Only compact long-term memory representations are exchanged, which are dimension-reduced and temporally aggregated abstractions of local experience. Therefore,* `SoHip` *provides intrinsic protection against direct data and model leakage.*

*Discussion.* Unlike parameter- or representation-sharing methods, `SoHip` decouples collaboration from model structure and data semantics. This design ensures that social knowledge exchange is achieved without exposing sensitive information, making `SoHip` particularly suitable for privacy-sensitive and heterogeneous environments.

Formal assumptions and proofs are provided in Appendix B.

# 6. Experiments

All experiments are implemented in PyTorch and conducted on a workstation equipped with NVIDIA RTX 3090 GPUs.

## 6.1. Experimental Setup

**Datasets and Data Partition.** We conduct experiments on two image classification benchmarks, CIFAR-100 with 100 classes [2] (Krizhevsky et al., 2009) and Tiny-ImageNet [3] (Chrabaszcz et al., 2017) with 200 classes. To simulate pathological non-IID data distributions, we adopt a label-skew partition strategy. For CIFAR-100, each agent is assigned data from 10 classes, while for Tiny-ImageNet, each agent is assigned data from 20 classes. Classes are randomly selected for each agent, resulting in highly heterogeneous local data distributions across agents.

**Models.** We evaluate `SoHip` under a heterogeneous model setting, where different agents employ convolutional neural networks with heterogeneous structures. This setup follows the common practice in heterogeneous collaborative learning (*e.g.*, `FedMRL`) and allows us to assess the robustness of memory-based social collaboration among agents, without requiring architectural alignment across agents.

**Baselines.** We compare `SoHip` with representative baselines covering different collaboration paradigms among heterogeneous agents. `Standalone` trains each agent independently without any collaboration. *Intermediate representation sharing* methods, including `FedProto` (Tan et al., 2022b), `FedSSA` (Yi et al., 2024c), and `FedRAL` (Yi et al., 2025a), enable collaboration by exchanging or aligning intermediate representations across agents. *Auxiliary homogeneous model sharing* methods, such as `FedKD` (Wu et al., 2022), `FedMRL` (Yi et al., 2024d), and `pFedES` (Yi et al., 2025b), introduce an additional homogeneous model as a knowledge transfer medium between heterogeneous agents. These baselines represent state-of-the-art approaches for collaborative learning under heterogeneity and provide a comprehensive comparison for evaluating `SoHip`.

**Evaluation Metric.** We report **average test accuracy** across all agents. After training, each agent evaluates its local model on the corresponding test set, and the overall performance is computed as $\text{Acc} = \frac{1}{N} \sum_{i=1}^N \text{Acc}_i$, where $\text{Acc}_i$ denotes the classification accuracy of agent $i$. This metric reflects the overall collaborative performance under model heterogeneity and non-IID data. All reported results are averaged over multiple runs with different random seeds.

---

[2] https://www.cs.toronto.edu/%7Ekriz/cifar.html
[3] http://cs231n.stanford.edu/tiny-imagenet-200.zip

*Table 1.* Average test accuracy (%) under the pathological label-skew partition with client participation rate $C = 10\%$ and varying number of clients $N$. Results are reported as mean $\pm$ standard deviation over three runs. Best results in each column are in bold.

| Method | CIFAR-100 | | | ImageNet | | |
|---|---|---|---|---|---|---|
| | $N$=100 | $N$=200 | $N$=300 | $N$=100 | $N$=200 | $N$=300 |
| Standalone | 53.59±0.48 | 47.35±0.52 | 42.92±0.61 | 35.36±0.57 | 28.29±0.63 | 25.94±0.69 |
| FedProto (Tan et al., 2022b) | 53.54±0.44 | 45.25±0.58 | 43.90±0.55 | 34.43±0.60 | 28.05±0.66 | 24.66±0.72 |
| FedSSA (Yi et al., 2024c) | 47.39±0.63 | 42.98±0.69 | 41.04±0.74 | 29.99±0.71 | 25.90±0.77 | 22.51±0.81 |
| FedRAL (Yi et al., 2025a) | 53.32±0.46 | 45.56±0.51 | 44.62±0.59 | 35.31±0.55 | 27.82±0.61 | 25.82±0.67 |
| FedKD (Wu et al., 2022) | 35.39±0.82 | 29.86±0.91 | 26.56±0.96 | 24.54±0.88 | 17.08±0.94 | 13.37±1.02 |
| FedMRL (Yi et al., 2024d) | 60.26±0.41 | 48.77±0.56 | 42.39±0.64 | 37.42±0.49 | 33.96±0.57 | 29.81±0.62 |
| pFedES (Yi et al., 2025b) | 50.02±0.59 | 46.71±0.62 | 42.44±0.68 | 36.83±0.54 | 28.96±0.66 | 23.71±0.73 |
| SoHip (Ours) | **63.33±0.36** | **54.33±0.42** | **50.10±0.48** | **46.20±0.41** | **36.12±0.46** | **34.30±0.51** |

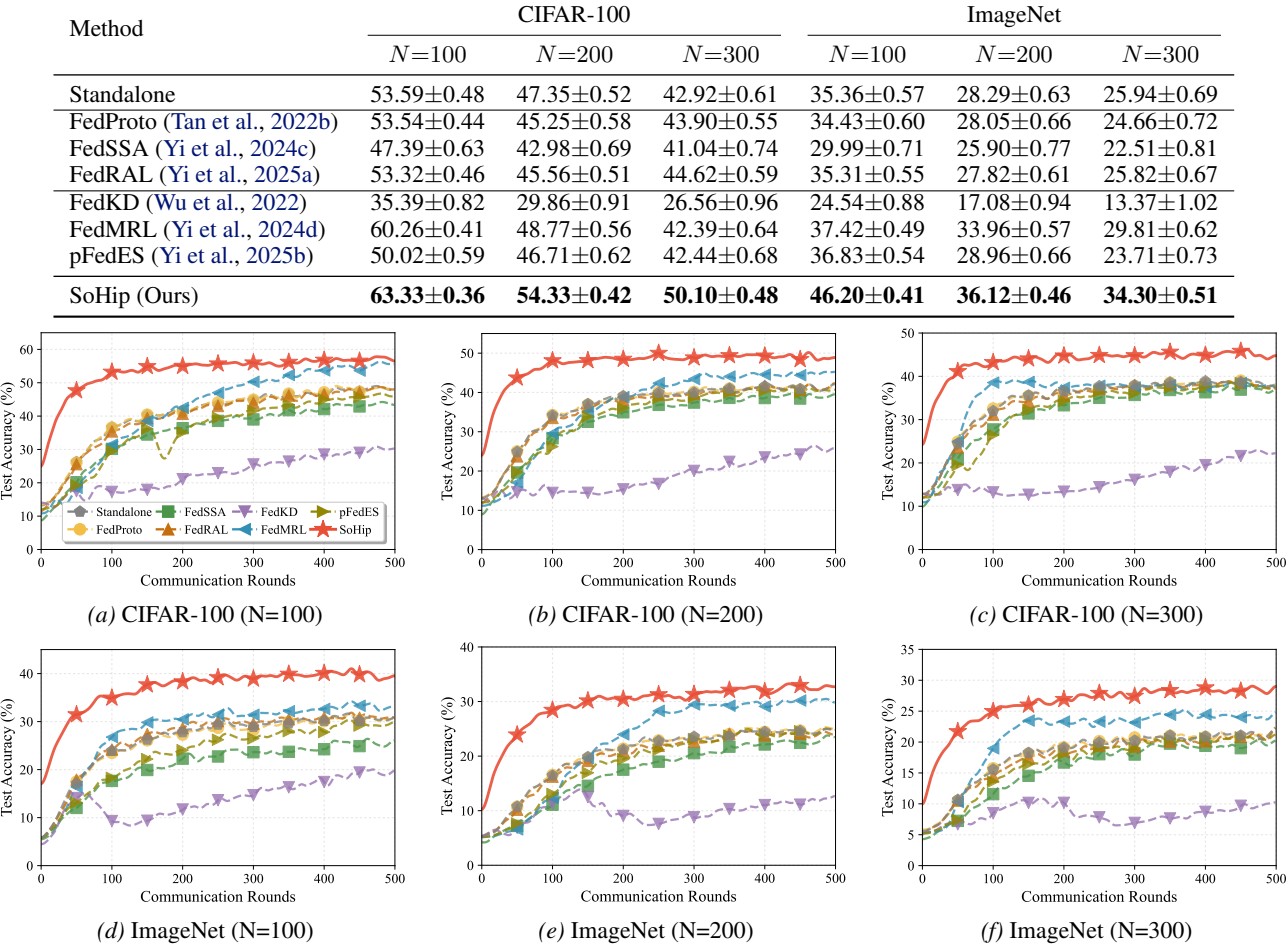

*(a)* CIFAR-100 (N=100)  *(b)* CIFAR-100 (N=200)  *(c)* CIFAR-100 (N=300)

*(d)* ImageNet (N=100)  *(e)* ImageNet (N=200)  *(f)* ImageNet (N=300)

*Figure 3.* Test accuracy curves under pathological label-skew settings ($C = 10\%$) on CIFAR-100 (top) and Tiny-ImageNet (bottom) with varying numbers of agents, where SoHip consistently achieves faster convergence and higher final accuracy across all settings.

**Hyperparameter Settings.** We consider the number $N = \{100, 200, 300\}$ of agents in all experiments, with a fixed participation rate of $C = 10\%$ per communication round. The total number of rounds is set to $T = 500$, which is sufficient to ensure convergence for all compared methods. All agents are optimized using SGD with a learning rate of $0.01$. The local batch size is set to $512$, and each agent performs 10 local epochs per round. Unless otherwise specified, all hyperparameters use these default settings.

### 6.2. Experimental Results

#### 6.2.1. OVERALL PERFORMANCE COMPARISON

**Performance comparison.** As shown in Table 1, SoHip achieves the highest test accuracy in all evaluated settings on both CIFAR-100 and Tiny-ImageNet. On CIFAR-100, SoHip improves the best competing method by up to +5.56% (54.33% → 48.77% at N=200) and maintains

clear advantages as the number of agents increases. On Tiny-ImageNet, the improvement is even more pronounced, reaching up to +8.78% (46.20% →37.42% at N=100), demonstrating the effectiveness of memory-based collaboration under more challenging fine-grained classification tasks. These consistent gains indicate that SoHip enables more effective knowledge sharing than parameter-, representation-, or auxiliary-model-based methods.

**Convergence behavior.** Figure 3 further illustrates the training dynamics under different agent numbers. SoHip converges significantly faster and reaches a higher accuracy plateau than all baselines across all settings. The accuracy gap emerges early in training and remains stable throughout communication rounds, suggesting that the proposed short-term memory abstraction and long-term memory consolidation allow agents to exploit shared knowledge more efficiently. In contrast, baseline methods exhibit slower

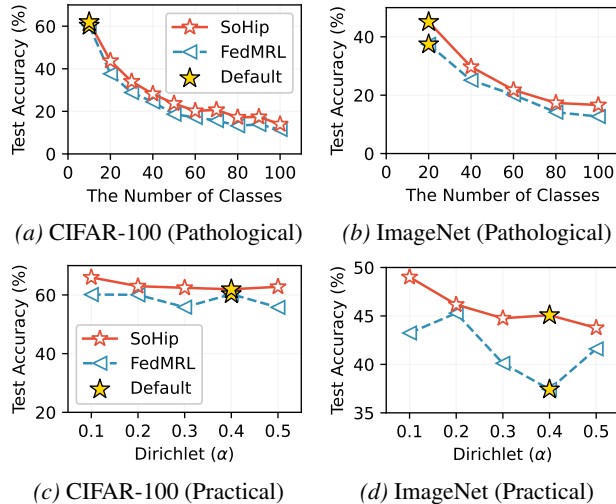

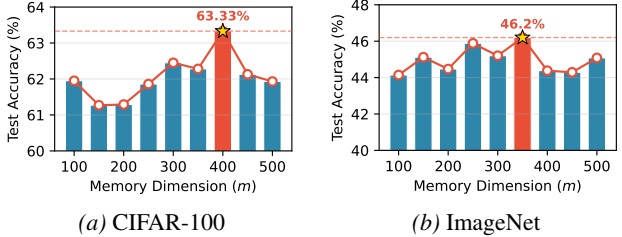

*Figure 5.* Impact of memory dimension $m$ on `SoHip`.

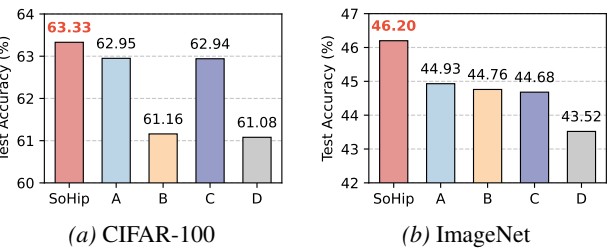

*Figure 6.* Ablation results of `SoHip`.

*Figure 4.* Impact of non-IID degree under pathological label-skew by varying classes per agent and practical label-skew by Dirichlet partition with concentration $\alpha$ ($\star$ is default in Table 1).

convergence and lower final performance, especially as the number of agents increases.

**Summary.** The above results demonstrate that `SoHip` not only achieves the highest final accuracy but also provides faster and more stable convergence, validating the advantage of memory-centric social machine learning under heterogeneous model and non-IID data conditions.

### 6.2.2. IMPACT OF NON-IID DEGREE.

Figure 4 reports the impact of data heterogeneity under two partition strategies. Across all settings, `SoHip` consistently outperforms `FedMRL`, indicating more effective knowledge transfer under heterogeneous data. As the number of classes per agent or the Dirichlet parameter $\alpha$ increases, the non-IID degree is reduced and the overall accuracy gradually decreases. This trend suggests that reduced data heterogeneity weakens inter-agent complementarity, limiting the benefit of collaborative memory sharing. In contrast, under stronger non-IID conditions, `SoHip` better exploits diverse and complementary local experience through memory exchange, resulting in superior performance than `FedMRL`.

### 6.2.3. IMPACT OF HYPERPARAMETER.

We investigate the impact of the memory dimension $m$, the only core hyperparameter in `SoHip`. Figure 5 shows that increasing the memory dimension $d_m$ initially improves performance, while overly large memory leads to slight degradation due to redundant or noisy information. Notably, `SoHip` consistently outperforms the strongest baseline `FedMRL` under all memory dimension settings. This verifies that `SoHip` is robust to memory dimension choices

and does not require careful hyperparameter tuning. [4]

### 6.2.4. ABLATION STUDY

We evaluate the contribution of each memory component in `SoHip` by progressively removing them. Variant **A** removes the importance gating in short-term memory abstraction, **B** discards the hippocampus-inspired consolidation and directly replaces new long-term memory with short-term memory, **C** removes collective long-term memory fusion, and **D** removes all memory modules. Figure 6 shows that the full `SoHip` consistently achieves the best performance on both datasets, while all ablated variants suffer performance degradation, with the largest drop observed in **D**. These results demonstrate that importance-aware short-term abstraction, hippocampus-inspired consolidation, and individual–collective memory fusion are necessary and jointly contribute to the effectiveness of `SoHip`.

## 7. Conclusion

This work proposes `SoHip`, a memory-centric social machine learning framework that enables effective collaboration among heterogeneous agents without sharing raw data or local model parameters. By abstracting, consolidating, and exchanging memory, `SoHip` provides a principled mechanism for social knowledge sharing under heterogeneity and privacy constraints. Theoretical analysis establishes convergence and privacy properties, and empirical results demonstrate consistent performance gains over existing heterogeneous federated learning methods.

---

[4]The impacts of agent participation rate and learning rate on `SoHip` are given in Appendix C.

## Acknowledgements

This work was supported in part by the National Natural Science Foundation of China under Grants U23B2049 and 22527901, the Accounting Centre of China Aviation under Grant 2025GFW-1232. We also thank the reviewers and area chairs for their valuable and constructive suggestions.

## Impact Statement

This paper presents work whose goal is to advance the field of Machine Learning. There are many potential societal consequences of our work, none which we feel must be specifically highlighted here.

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

# A. Key Notations.

Table 2 summarizes the key notations used throughout the paper.

*Table 2.* Summary of notations used in `SoHip`.

| Notation | Description |
| --- | --- |
| $N$ | Total number of agents (clients). |
| $i$ | Index of an agent, $i \in \{1, \ldots, N\}$. |
| $t$ | Communication round index. |
| $\mathcal{D}_i$ | Local private dataset of agent $i$. |
| $\mathcal{F}_i$ | Heterogeneous feature extractor of agent $i$. |
| $\mathcal{H}_i$ | Local classifier (prediction head) of agent $i$. |
| $\mathcal{E}_i$ | Local memory encoder (linear projection to memory space). |
| $\mathcal{R}_i$ | Local memory decoder (linear projection to feature space). |
| $\mathcal{B}_i^t$ | Mini-batch sampled from $\mathcal{D}_i$ at round $t$. |
| $\mathbf{Z}_i^t \in \mathbb{R}^{B_i^t \times d_i}$ | Latent representations extracted from $\mathcal{B}_i^t$ by $\mathcal{F}_i$. |
| $d_i$ | Feature dimension of agent $i$'s local model. |
| $m$ | Shared memory dimension across all agents. |
| $\bar{\mathbf{z}}_i^t \in \mathbb{R}^m$ | Batch-averaged encoded representation at round $t$. |
| $\boldsymbol{\alpha}_i^{\mathrm{S},t} \in (0,1)^m$ | Short-term memory gating vector controlling importance of recent observations. |
| $\mathbf{M}_i^{\mathrm{S},t} \in \mathbb{R}^m$ | Individual short-term memory of agent $i$ at round $t$. |
| $\mathbf{M}_i^{\mathrm{L},t} \in \mathbb{R}^m$ | Individual long-term memory of agent $i$ after consolidation at round $t$. |
| $\mathbf{M}^{\mathrm{L},t} \in \mathbb{R}^m$ | Collective long-term memory aggregated by the server at round $t$. |
| $\boldsymbol{\alpha}_i^{\mathrm{in},t} \in (0,1)^m$ | Input gate controlling incorporation of short-term memory. |
| $\boldsymbol{\alpha}_i^{\mathrm{f},t} \in (0,1)^m$ | Forget gate controlling retention of historical long-term memory. |
| $\boldsymbol{\alpha}_i^{\mathrm{o},t} \in (0,1)^m$ | Output gate modulating consolidated long-term memory strength. |
| $\boldsymbol{\alpha}_i^{\mathrm{G},t} \in (0,1)^m$ | Fusion gate controlling absorption of collective memory. |
| $\mathbf{M}_i^t \in \mathbb{R}^m$ | Complete memory of agent $i$ after individual–collective fusion. |
| $\tilde{\mathbf{m}}_i^t \in \mathbb{R}^{d_i}$ | Decoded memory projected back to the feature space. |
| $\hat{\mathbf{Z}}_i^t$ | Memory-enhanced representations for prediction. |
| $\hat{\mathbf{Y}}_i^t$ | Prediction outputs of agent $i$ at round $t$. |
| $\ell(\cdot, \cdot)$ | Local prediction loss function (e.g., cross-entropy). |
| $\mathcal{L}$ | Global objective function aggregating all local losses. |
| $\mathcal{S}_t$ | Set of participating agents at round $t$. |
| $p_i$ | Aggregation weight of agent $i$ (e.g., proportional to $|\mathcal{D}_i|$). |
| $C$ | Client participation rate per communication round. |
| $\sigma(\cdot)$ | Sigmoid activation function. |

## B. Theoretical Analysis

This appendix provides detailed analysis supporting the convergence and privacy claims presented in Section 5. We follow standard assumptions in federated and distributed optimization and adapt them to the memory-based social machine learning setting of `SoHip`.

### B.1. Preliminaries and Assumptions

We consider the global objective

$$f(\theta) = \sum_{i=1}^{N} p_i f_i(\theta), \tag{17}$$

where $f_i(\theta) := \mathbb{E}_{(\mathbf{x},y)\sim\mathcal{D}_i}\big[\ell(\mathcal{H}_i(\mathcal{F}_i(\mathbf{x})), y)\big]$ denotes the local objective of agent $i$, and $p_i \geq 0$, $\sum_i p_i = 1$.

**Assumption 1 (Smoothness).** Each local objective $f_i$ is $L$-smooth, i.e.,

$$\|\nabla f_i(\theta) - \nabla f_i(\theta')\| \leq L\|\theta - \theta'\|, \quad \forall \theta, \theta'.$$

**Assumption 2 (Unbiased Stochastic Gradients).** Each agent computes stochastic gradients $\nabla f_i(\theta; \xi)$ such that

$$\mathbb{E}_\xi[\nabla f_i(\theta; \xi)] = \nabla f_i(\theta),$$

with bounded variance $\mathbb{E}_\xi\|\nabla f_i(\theta; \xi) - \nabla f_i(\theta)\|^2 \leq \sigma^2$.

**Assumption 3 (Bounded Heterogeneity).** There exists $\Delta_{\text{het}} \geq 0$ such that

$$\sum_{i=1}^{N} p_i\|\nabla f_i(\theta) - \nabla f(\theta)\|^2 \leq \Delta_{\text{het}}, \quad \forall \theta.$$

These assumptions are standard in nonconvex federated optimization and hold independently of the memory abstraction mechanism.

### B.2. Convergence Analysis

We analyze the effect of memory-based collaboration on the optimization dynamics of `SoHip`.

In `SoHip`, local model updates are performed using memory-enhanced representations, where the memory modules (short-term abstraction, consolidation, and fusion) act as deterministic, differentiable transformations parameterized by lightweight neural networks. Importantly, memory exchange does not introduce additional stochasticity into gradient estimation.

Let $\theta^t$ denote the collection of local model parameters at communication round $t$. Following standard analysis for stochastic gradient methods, we have

$$\mathbb{E}[f(\theta^{t+1})] \leq \mathbb{E}[f(\theta^t)] - \eta\mathbb{E}\|\nabla f(\theta^t)\|^2 + \frac{L\eta^2}{2}\mathbb{E}\|g^t\|^2, \tag{18}$$

where $g^t$ denotes the aggregated stochastic gradient and $\eta$ is the learning rate.

Using Assumptions 1–3 and standard variance decomposition, the gradient norm can be bounded as

$$\mathbb{E}\|g^t\|^2 \leq 2\mathbb{E}\|\nabla f(\theta^t)\|^2 + 2(\sigma^2 + \Delta_{\text{het}}). \tag{19}$$

Substituting the bound and telescoping over $t = 0, \ldots, T-1$ yields

$$\frac{1}{T}\sum_{t=0}^{T-1}\mathbb{E}\|\nabla f(\theta^t)\|^2 \leq \mathcal{O}\Big(\frac{1}{\eta T}\Big) + \mathcal{O}(\eta\sigma^2) + \mathcal{O}(\eta\Delta_{\text{het}}). \tag{20}$$

Choosing $\eta = \mathcal{O}(1/\sqrt{T})$ leads to the convergence rate stated in Theorem 5.1:

$$\frac{1}{T}\sum_{t=0}^{T-1}\mathbb{E}\|\nabla f(\theta^t)\|^2 = \mathcal{O}\Big(\frac{1}{\sqrt{T}}\Big) + \mathcal{O}(\Delta_{\text{het}}).$$

**Remarks.** The key observation is that memory abstraction, consolidation, and fusion do not alter the fundamental optimization structure. They act as bounded, differentiable transformations applied consistently across iterations. Therefore, `SoHip` preserves the convergence guarantees of federated optimization while improving empirical performance through structured knowledge sharing.

### B.3. Privacy Preservation Analysis

We analyze the privacy properties of `SoHip` from an architectural perspective.

**Observation 1 (No Raw Data Sharing).** At no stage does `SoHip` transmit raw samples $\mathbf{x}$ or labels $y$. All operations involving raw data (feature extraction, memory abstraction, and prediction) are performed locally on-device.

**Observation 2 (No Model Parameter Sharing).** Local model parameters $\mathcal{F}_i$, $\mathcal{H}_i$, as well as memory encoders and decoders, are never transmitted. Only memory vectors $\mathbf{M}_i^{\mathrm{L},t} \in \mathbb{R}^m$ are uploaded to the server.

**Observation 3 (Abstracted and Non-Invertible Memory).** The transmitted memory is: (i) dimension-reduced ($m \ll d_i$); (ii) gated and nonlinear; (iii) temporally aggregated across batches and rounds. These properties make direct reconstruction of local data or model parameters ill-posed.

**Proposition.** Given only the transmitted long-term memory $\mathbf{M}_i^{\mathrm{L},t}$, recovering the original local data or model parameters is underdetermined without access to private encoders, gating functions, and historical context.

**Discussion.** Unlike differential privacy mechanisms, `SoHip` provides *structural privacy* by design. Memory acts as a high-level abstraction of experience, not a carrier of raw information. This makes `SoHip` compatible with existing privacy-enhancing techniques (e.g., DP or secure aggregation), while already offering strong intrinsic protection against direct information leakage.

## C. More Experimental Results

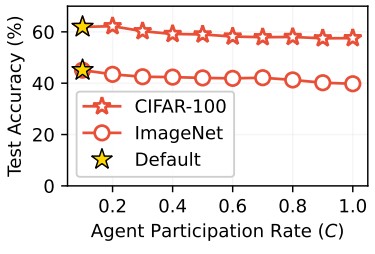

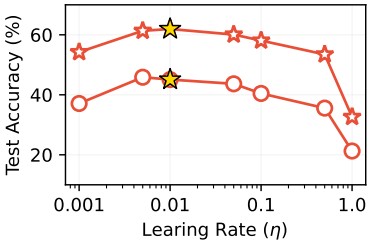

*(a)* Impact of agent participation rate.
*(b)* Impact of learning rate.

*Figure 7.* Sensitivity analysis of `SoHip` with respect to agent participation rate and learning rate.

**Impact of Agent Participation Rate.** Figure 7a reports the impact of the agent participation rate on `SoHip` under CIFAR-100 and ImageNet. As the participation fraction increases, the test accuracy of `SoHip` shows a mild decreasing trend on both datasets. This behavior can be attributed to the fact that higher participation introduces more heterogeneous and potentially conflicting local updates within each round, which increases the difficulty of consolidating consistent long-term memory. Nevertheless, `SoHip` remains stable across a wide range of participation rates, and the default setting achieves a favorable balance between performance and communication efficiency, demonstrating the robustness of memory-based collaboration under varying participation levels.

**Impact of Learning Rate.** Figure 7b illustrates the impact of the learning rate on `SoHip` across CIFAR-100 and ImageNet. The performance exhibits a clear unimodal trend: very small learning rates lead to slow and suboptimal convergence, while excessively large learning rates cause unstable updates and significant performance degradation. An intermediate learning rate (default setting) consistently yields the best accuracy on both datasets, indicating a good balance between convergence speed and training stability. These results suggest that `SoHip` is relatively robust to learning rate choices within a reasonable range, while extreme settings may hinder effective memory consolidation and fusion.

