# OpenReview forum: "Social Hippocampus Memory Learning"
_ICML.cc/2026/Conference — ICML 2026 regular_

### Official Review · Reviewer_WaAU · 2026-03-07

**Soundness:** 2
**Presentation:** 2
**Significance:** 2
**Originality:** 2
**Overall Recommendation:** 4
**Confidence:** 3

**Summary:**

This paper introduces a memory-centric social machine learning framework, SoHip. Unlike traditional federated learning, in which a central server aggregates local trained models from clients, the proposed SoHip collects each client's individual long-term memory. Specifically, long-term memory is converted from the abstract knowledge in short-term memory by a hippocampal module. Extensive experiments demonstrate the effectiveness of SoHip.

**Compliance With Llm Reviewing Policy:**

Affirmed.

**Final Justification:**

Most of my concerns have been addressed. I have raised my score to 4.

**Key Questions For Authors:**

See **Weaknesses**.

**Limitations:**

See **Weaknesses**.

**Strengths And Weaknesses:**

**Strengths:**
1. The paper is well organized and generally easy to follow.
2. This work studies a scenario that is meaningfully different from standard federated learning. This memory-centric perspective is interesting and could be of value to the heterogeneous federated learning community.

**Weaknesses:**
1. The privacy claim is not sufficiently justified. There is no formal differential privacy (DP) theorem or privacy accounting in the paper. Theorem 5.2 mainly states that raw data, local model parameters, intermediate features, and prediction outputs are not transmitted. Appendix B further describes this as structural privacy by design, rather than a differential privacy guarantee. In my view, this is not sufficient to support a strong privacy-preservation claim.
2. The convergence result is also somewhat limited. Theorem 5.1 offers a rate that contains a non-zero term $\Delta_{het}$. Although the authors argue that this term is unavoidable in heterogeneous settings, several recent works introduce analytical frameworks [1, 2] that aim to make the overall convergence error approach zero even under dynamic or model-heterogeneous settings. This limitation should be discussed more carefully.
3. The experimental setup is not described clearly enough. In particular, the paper only states that clients use convolutional neural networks with heterogeneous structures, but it does not specify the exact architectures. Even if the CNNs are designed by the authors, the architectural details should be reported. It would also strengthen the experimental section by including more standard backbones, such as ResNet variants, to make the comparison more convincing.
4. The empirical comparison does not include several important baselines for heterogeneous-model federated learning, such as HeteroFL [3] and FedRolex [4]. These baselines address partial model aggregation, but they are not included in the paper's discussion.
5. The paper introduces a new memory-based terminology, but the connection to experience replay in federated learning is not discussed clearly enough. The method explicitly maintains short-term and long-term memory, updates them over rounds, and reuses them during local training and server aggregation. Because of this, the proposed idea appears closely related to experience-replay methods [5, 6]. The paper should clarify more explicitly how SoHip differs conceptually and technically from experience replay, especially since both involve retaining and reusing locally derived information.

**References:**
[1] Every parameter matters: Ensuring the convergence of federated learning with dynamic heterogeneous models reduction
[2] FIARSE: Model-Heterogeneous Federated Learning via Importance-Aware Submodel Extraction
[3] HeteroFL: Computation and Communication Efficient Federated Learning for Heterogeneous Clients
[4] FedRolex: Model-Heterogeneous Federated Learning with Rolling Sub-Model Extraction
[5] FedER: Federated Learning through Experience Replay and privacy-preserving data synthesis
[6] Experience Replay as an Effective Strategy for Optimizing Decentralized Federated Learning

---

> ### Author Rebuttal · Authors · 2026-03-31
>
> We thank the reviewer for the positive feedback on the organization and clarity of the paper, as well as for recognizing the value of the memory-centric perspective in heterogeneous social machine learning. We also appreciate the constructive comments and address them below.
>
> ---
>
> **W1:** We thank the reviewer for this comment.
> Please refer to our response to **R2-W2**, where we clarify that SoHip does not aim to provide formal privacy guarantees (e.g., differential privacy), but rather a **privacy-friendly design**. In addition, since SoHip only exchanges abstracted memory representations, it is **compatible with standard privacy-enhancing techniques** such as differential privacy or secure aggregation. We will revise the wording accordingly to avoid over-claiming.
>
> ---
>
> **W2:** We thank the reviewer for this insightful comment and for pointing out relevant works.
> We agree that more refined analyses can reduce or eliminate the residual error under specific settings. Our goal is to provide a **general convergence characterization under model and data heterogeneity**, where the residual term reflects inherent discrepancies across clients rather than analysis limitations.
>
> We note that many **existing works achieving vanishing error rely on additional assumptions or mechanisms** (e.g., structured model alignment or constrained updates), whereas our framework considers a more **general setting with fully heterogeneous models and memory-based collaboration**. We will clarify this positioning in the revision.
>
> ---
>
> **W3:** We thank the reviewer for this helpful comment.
> We agree that architectural details should be more clearly specified. In our experiments, heterogeneous CNNs differ in kernel sizes, network depths, and linear dimensions, following **FedMRL (NeurIPS’24, Line 297)**. We will include detailed descriptions in the revision.
>
> In addition, please refer to **R2-W1**, where we include experiments with standard backbones (e.g., **ResNet variants**) under stronger heterogeneity, further demonstrating strong performance and broad applicability.
>
> ---
>
> **W4:** We thank the reviewer for this suggestion and for pointing out relevant methods.
> We agree that **HeteroFL** and **FedRolex** are important works. However, these methods fundamentally **rely on structured model alignment**, where each client must be a **sub-network of a shared global model** for parameter aggregation.
>
> In contrast, we consider **fully heterogeneous models without shared parameter space**, where such alignment does not exist. Therefore, these methods are **not directly applicable or comparable** to our setting. We will clarify this distinction in the revision.
>
> ---
>
> **W5:** We thank the reviewer for this insightful comment.
> While both SoHip and experience replay involve retaining historical information, they differ fundamentally:
>
> (1) Experience replay stores and reuses **data samples or synthetic data**, whereas SoHip exchanges **abstracted memory representations** of local experience.
> (2) Experience replay is primarily a **local mechanism**, while SoHip enables **cross-client memory aggregation and evolution** across rounds.
>
> Conceptually, replay operates at the **data level**, while SoHip operates at the **memory (knowledge) level**, decoupled from raw data and model structure. Therefore, SoHip represents a **memory-centric collaboration paradigm**, rather than a variant of data replay. We will clarify this distinction in the revision.
>
> ---
>
> **Overall Response:** We thank the reviewer again. This work introduces a **memory-centric perspective for social machine learning under fully heterogeneous agents**, where collaboration is achieved through abstracted memory rather than parameter, representation, or data sharing.
>
> Together with **consistent empirical improvements, broader experimental evidence, and clarified theoretical and privacy positioning**, we believe this work provides a meaningful and practical direction for heterogeneous social machine learning including federated learning and may be of interest to a broad ICML audience.

---

> > ### Author Rebuttal · Reviewer_WaAU · 2026-04-02
> >
> > Thank you for your response. Most of my concerns have been addressed. I will raise my score to 4.

---

> > > ### Author Response · Authors · 2026-04-03
> > >
> > > Thank you for your thoughtful review and for taking the time to consider our rebuttal. We truly appreciate your recognition that the concerns have been addressed, and thank you for updating your score.

---

### Official Review · Reviewer_YNdp · 2026-03-11

**Soundness:** 3
**Presentation:** 3
**Significance:** 2
**Originality:** 2
**Overall Recommendation:** 4
**Confidence:** 4

**Summary:**

This paper proposes SoHip, a memory-centric framework for heterogeneous federated/social machine learning. Instead of exchanging raw data, model parameters, or predictions, each client builds short-term memory from local features, consolidates it into long-term memory through a hippocampus-inspired module, and shares only the long-term memory with the server for aggregation. The paper argues that this design is better suited for heterogeneous collaborative learning and may offer improved privacy. Experiments on CIFAR-100 and Tiny-ImageNet with heterogeneous CNN clients show consistent gains over several heterogeneous FL baselines.

**Compliance With Llm Reviewing Policy:**

Affirmed.

**Final Justification:**

The authors' rebuttal effectively resolved my questions, and I am happy to update my score as my concerns have been positively addressed.

**Key Questions For Authors:**

1.How much of the improvement comes from the specific SoHip memory design, compared with simply exchanging a compact latent vector of the same size?
2.Can the authors either provide a stronger privacy evaluation or tone down the privacy claims?
3.Since the paper is framed broadly, can the authors provide evidence beyond the current image-classification setting?
4.How sensitive is SoHip to the design choices of the memory dimension, gating functions, and fusion strategy?
5.What is the communication and computation overhead introduced by the proposed memory modules?

**Limitations:**

The paper does not adequately discuss its limitations. The current impact statement says that there are no societal consequences that need to be highlighted, and the paper also does not sufficiently acknowledge technical limitations such as the narrow benchmark scope, lack of privacy attack evaluation, and limited evidence for broad generalization beyond image classification. A stronger limitations section would improve the paper.

**Strengths And Weaknesses:**

Strengths:
The paper studies an important and practically relevant setting: collaborative learning under both data heterogeneity and model heterogeneity. This is a meaningful problem, since many existing federated learning methods are still more naturally suited to homogeneous model settings. The central idea of using compact exchanged memory, rather than directly sharing model parameters, intermediate features, or predictions, is interesting and reasonably motivated. In particular, the paper offers a clean perspective on how heterogeneous clients might collaborate through a more architecture-agnostic information carrier.
Another strength is that the method is presented in a fairly clear and modular way. The overall pipeline—short-term memory construction, hippocampus-inspired consolidation, local-global fusion, and server-side aggregation—is easy to follow, and the paper does a good job of explaining the role of each component. The presentation is generally solid, with a logical flow from motivation to method to experiments.
Empirically, the reported results are consistently positive within the tested setup. SoHip outperforms the compared heterogeneous FL baselines across the reported CIFAR-100 and Tiny-ImageNet settings, and the gains are not isolated to a single configuration. The ablation study is also useful: it suggests that the proposed memory-related modules each make a contribution, rather than the gains coming from only one dominant component. Overall, the paper is well executed and presents a coherent story.

Weaknesses:
My main concern is that the paper’s claims are broader than the evidence currently supports. While the paper is framed as a general framework for heterogeneous social/federated machine learning, the empirical evaluation is limited to image classification on two datasets, using a relatively narrow family of heterogeneous CNN architectures. This makes it difficult to judge whether the proposed idea is truly robust across broader settings, such as other modalities, tasks, or stronger forms of architectural heterogeneity. As a result, the scope of the experimental evidence feels narrower than the framing of the contribution.

I also find the privacy claim somewhat under-supported in its current form. The method is clearly more privacy-friendly at an architectural level, since it avoids directly exchanging raw data, model parameters, and outputs. However, this is not the same as providing a formal privacy guarantee, and the paper does not include empirical privacy analysis such as reconstruction, inversion, or leakage-related attacks. Given that privacy is part of the motivation, I think the paper should either provide stronger evidence here or state the claim more carefully.

A further concern is that the algorithmic novelty feels somewhat moderate relative to the conceptual framing. The paper’s narrative around hippocampus-inspired memory learning is appealing, but the implemented mechanism appears closer to a gated latent-memory exchange/update design than to a fundamentally new learning principle. That does not make the paper uninteresting, but it does make the technical contribution feel more incremental than the high-level framing may suggest.

Finally, I think the experimental section would be stronger with more diagnostic comparisons. In particular, it would be useful to compare against simpler latent-sharing or prototype-sharing baselines with matched communication budgets. Without such controls, it is hard to tell how much of the gain comes specifically from the proposed memory design, rather than from the more general effect of adding an exchanged compact latent representation.

Overall, I find the paper interesting, clearly written, and empirically promising within the presented setup. However, at the current stage, I am not fully convinced that the submission provides sufficiently broad evidence or sufficiently strong technical novelty for ICML acceptance.

---

> ### Author Rebuttal · Authors · 2026-03-31
>
> We sincerely thank the reviewer for the thoughtful and constructive feedback. We are encouraged that the reviewer finds the problem setting meaningful, the idea of memory-based collaboration interesting, and the empirical results consistently positive.
>
> ---
>
> **W1:** We thank the reviewer for this suggestion.
>
> Following the feedback, we add three additional experiments under broader settings:
> (1) **High model heterogeneity:** CIFAR-100 with diverse CNNs and ResNet variants;
> (2) **NLP task:** Stack Overflow next-word prediction with heterogeneous LSTMs;
> (3) **Healthcare task:** FLamby benchmark with real-world federated data.
>
> | Method | High Heterogeneity | NLP | Healthcare |
> |--------|------------------|-----|------------|
> | Standalone | 45.58 | 16.35 | 65.83 |
> | FedProto | 48.56 | 23.28 | 67.62 |
> | FedSSA | 47.39 | 22.95 | 66.90 |
> | FedRAL | 48.92 | 23.12 | 67.54 |
> | FedKD | 32.43 | 17.53 | 62.48 |
> | FedMRL | 49.36 | 23.86 | 69.84 |
> | pFedES | 45.97 | 21.74 | 66.88 |
> | **SoHip** | **50.10** | **24.73** | **71.39** |
>
> SoHip consistently achieves the best performance, demonstrating strong generalization across modalities, tasks, and heterogeneous models. We will incorporate these results in the revision.
>
> ---
>
> **W2:** We thank the reviewer for this comment.
> SoHip does not provide formal privacy guarantees (e.g., DP), and we will revise the wording to avoid over-claiming. It is **privacy-friendly by design**, avoiding sharing data, parameters, features, and outputs, and only exchanging compact memory representations. This design is orthogonal to and **compatible** with techniques such as differential privacy or secure aggregation. We will clarify this positioning and include empirical privacy analysis as future work.
>
> ---
>
> **W3:** We thank the reviewer for this insightful comment.
> While components are lightweight, the novelty lies in a **memory-centric collaboration paradigm**, fundamentally different from latent or representation sharing. The exchanged memory is a **temporally evolving knowledge abstraction**, refined across rounds and filtered via gating, enabling structured accumulation of memory. This enables **memory-level collaboration**, beyond parameter- or representation-level interaction.
>
> Ablations show that removing key memory components degrades performance, confirming that gains come from **memory abstraction and evolution**, not simple latent exchange. We will clarify this in the revision.
>
> ---
>
> **W4:** We thank the reviewer for this suggestion.
> Our evaluation already includes representative baselines such as **FedProto** (prototype-sharing) and **FedRAL** (latent-sharing). SoHip consistently outperforms them and converges faster (Fig. 3), indicating that gains are not due to low-dimensional exchange alone.
>
> Ablations further show that removing memory components degrades performance, confirming that improvements stem from the **memory abstraction and evolution mechanism**. We will clarify this in the revision.
>
> ---
>
> **Overall Response:** We thank the reviewer again. We have **strengthened empirical evidence under broader settings** and clarified that the contribution lies in a **memory-centric collaboration paradigm**. We hope these address concerns on scope and novelty.
>
> ---
>
> **Q1:** We thank the reviewer. In ablation, **Variant D** (simple memory exchange) achieves 61.08%/43.52% on CIFAR-100/ImageNet, outperforming FedRAL (53.32%/35.31%) and FedProto (53.54%/34.43%). The full SoHip further improves to 63.33%/46.20%, showing gains come from **memory abstraction and evolution**, not compact exchange.
>
> **Q2:** Please refer to **W2**.
>
> **Q3:** Please refer to **W1**.
>
> **Q4:** Fig. 5 shows SoHip is stable across memory dimensions. Ablation (Fig. 6) confirms the effectiveness of gating and fusion while maintaining robustness.
>
> **Q5:** The overhead is **lightweight**: communication involves low-dimensional memory ($m \ll d_i$), and computation adds only simple linear and gating operations.
>
> ---
>
> **Limitations:** We thank the reviewer for this suggestion. The concerns raised have been addressed in our rebuttal. In addition, we will include a brief limitations discussion in the revision, mainly noting that SoHip introduces several lightweight memory-related modules (e.g., gating and fusion), which may bring additional design choices in practice.

---

> > ### Author Rebuttal · Reviewer_YNdp · 2026-04-01
> >
> > I thank the authors for their dedicated efforts during the rebuttal period. Most of my concerns have been effectively addressed; therefore, I am pleased to raise my score to 4.
> >
> > Furthermore, as the paradigm shifts from traditional Federated Learning (FL) to Federated Large Language Models (FedLLM), I strongly encourage the authors to explore extending the current framework to LLM-based scenarios. Such an extension would significantly broaden the impact and relevance of this work in the current LLM era.

---

> > > ### Author Response · Authors · 2026-04-02
> > >
> > > Thank you for the positive feedback and for carefully considering our rebuttal. We truly appreciate your recognition that the main concerns have been addressed.
> > >
> > > We also appreciate your insightful suggestion on extending SoHip to LLM-based scenarios. We agree that this is a promising direction, and we will explore adapting the memory-centric collaboration mechanism to federated LLM settings in future work.

---

### Official Review · Reviewer_2heC · 2026-03-21

**Soundness:** 4
**Presentation:** 3
**Significance:** 3
**Originality:** 4
**Overall Recommendation:** 6
**Confidence:** 5

**Summary:**

This paper studies social machine learning under agents with heterogeneous models and proposes a novel SoHip framework to enable collaboration among heterogeneous agents via memory sharing. In SoHip, each client first abstracts individual short-term memory, and then transforms it into individual long-term memory through an interesting hippocampus-inspired memory consolidation module, which is used to aggregate on the server to obtain the shared collective memory. Each client fuses the individual and collective long-term memory as the enhanced memory to improve model performance. Since only sharing memory, the private data and model within each client is effectively preserved, and the communication efficiency is also improved.

**Compliance With Llm Reviewing Policy:**

Affirmed.

**Key Questions For Authors:**

Can a more expressive gating design improve the importance scoring mechanism?

Can more advanced aggregation rules better exploit memory heterogeneity?

**Limitations:**

The importance scoring is simple. It uses a linear gate, and more expressive gating designs are not explored.

The memory aggregation is conventional.It relies on weighted averaging, and better aggregation strategies may exist.

**Strengths And Weaknesses:**

Strengths:
1.	This paper explores a novel and popular machine learning paradigm – social machine learning, especially under agents with model heterogeneity.
2.	To bridges the latent privacy leakage and high costs, this paper proposes an innovative Social Hippocampus Memory Learning (SoHip) framework, enables information exchange through memory sharing instead of traditional model parameter sharing.
3.	The three key designs in SoHip (Individual Short-Term Memory Abstraction, Hippocampus-Inspired Memory Consolidation, and Individual–Collective Memory Fusion) achieves memory-enhanced learning through abstracting, consolidating, and merging individual and collective memory, which is a complete memory split process and an interesting schema to achieve memory exchange.
4.	This paper provides strict theoretical proof and extensive experiments to demonstrate the effectiveness of the proposed SoHip algorithm.

Weaknesses：
1.	This paper calculates important scores through a linear gate, the author should explore other gate structures, such as a linear gate with multiple layers.
2.	This paper aggregates individual long-term memory through weighted averaging, which is often used in the traditional FL algorithms, such as FedAvg, whether other alternative aggregation rules are more effective?

---

> ### Author Rebuttal · Authors · 2026-03-31
>
> We sincerely thank the reviewer for the positive evaluation and constructive suggestions. We are encouraged that the reviewer recognizes the novelty and effectiveness of our memory-centric social machine learning framework for
> heterogeneous agents.
>
> ---
>
> **W1&Q1:** We agree that more expressive gates (e.g., deeper or nonlinear gating networks) are worth exploring. Our current design intentionally uses lightweight linear gates as a **minimal instantiation**, so that the improvements can be attributed to the proposed **memory-sharing paradigm** rather than architectural complexity. Even under this simple setting, SoHip consistently achieves superior performance over strong baselines, suggesting that the key benefit comes from memory abstraction and consolidation rather than gate expressiveness. We will clarify this motivation and include discussion of richer gating designs as future extensions.
>
> ---
>
> **W2&Q2:** We also agree that weighted averaging is not the only possible aggregation rule. We adopt it as a simple and stable baseline to isolate the effect of **memory-level collaboration** and ensure fair comparison with standard FL pipelines. Notably, SoHip operates at a different abstraction level by exchanging long-term memory rather than parameters, making it naturally compatible with more advanced aggregation strategies. We will add this discussion and highlight improved aggregation as a promising direction.

---

> > ### Author Rebuttal · Reviewer_2heC · 2026-04-01
> >
> > Thanks for the clarifications. The authors have addressed my concerns clearly. In my view, the paper introduces a novel and well-motivated memory-centric paradigm for social machine learning under heterogeneous agents, where collaboration is achieved through memory-centric knowledge sharing, offering a perspective that is clearly distinct from existing parameter- or representation-sharing approaches. The method is technically sound, and both the empirical and theoretical results are convincing. Given its conceptual contribution and practical relevance, I believe this work could be of interest to a broad ICML audience, so I will increase my score accordingly.

---

> > > ### Author Response · Authors · 2026-04-01
> > >
> > > Thank you for the positive feedback and for the thoughtful evaluation of our rebuttal.
> > > We appreciate your recognition of the memory-centric perspective and the overall contribution of our work.
> > > We are glad that the clarifications were helpful, and thank you for your support.

---

### Review · Ethics_Reviewer_PM3v · 2026-03-27

**Recommendation:** No remediation action needed

**Ethics Issue:**

Reviewer YNdp argued that the paper lacks a dedicated limitations section that would address potential societal consequences of the proposed method. While this is certainly an important concern that the authors and reviewers can discuss during the rebuttal period, I don't see this as a fundamental research integrity or ethics issue that needs to be resolved by further escalation up the ethics reviewing chain.

---

### Decision · Program_Chairs · 2026-04-30

**Decision:**

Accept (regular)

**Comment:**

The reviewers are in agreement on acceptance (6/4/4). All reviewers recognize the novelty of the memory-centric collaboration paradigm for heterogeneous federated/social machine learning.

Key concerns included limited benchmark scope, insufficient privacy justification, and moderate algorithmic novelty beyond the conceptual framing. The authors addressed these effectively by adding experiments.
I recommend acceptance.